# Enhanced Anticancer Activity of Atractylodin-Loaded Poly(lactic-co-glycolic Acid) Nanoparticles Against Cholangiocarcinoma

**DOI:** 10.3390/polym17152151

**Published:** 2025-08-06

**Authors:** Tullayakorn Plengsuriyakarn, Luxsana Panrit, Kesara Na-Bangchang

**Affiliations:** 1Center of Excellence in Molecular Biology and Pharmacology of Malaria and Cholangiocarcinoma, Chulabhorn International College of Medicine, Thammasat University, Pathum Thani 12120, Thailand; tulpleng@tu.ac.th; 2Faculty of Allied Health Science, Pathum Thani University, Pathum Thani 12000, Thailand; pluxana@tu.ac.th; 3The Office of Advanced Science and Technology, Thammasat University, Pathum Thani 12120, Thailand

**Keywords:** atractylodin, PLGA nanoparticle, cytotoxicity, cell invasion, apoptosis, genotoxicity, anti-cholangiocarcinoma activity

## Abstract

Cholangiocarcinoma (CCA) is highly prevalent in the Greater Mekong sub-region, especially northeastern Thailand, where infection with the liver fluke Opisthorchis viverrini is a major etiological factor. Limited therapeutic options and the absence of reliable early diagnosis tools impede effective disease control. *Atractylodes lancea* (Thunb.) DC.—long used in Thai and East Asian medicine, contains atractylodin (ATD), a potent bioactive compound with anticancer potential. Here, we developed ATD-loaded poly(lactic co-glycolic acid) nanoparticles (ATD PLGA NPs) and evaluated their antitumor efficacy against CCA. The formulated nanoparticles had a mean diameter of 229.8 nm, an encapsulation efficiency of 83%, and exhibited biphasic, sustained release, reaching a cumulative release of 92% within seven days. In vitro, ATD-PLGA NPs selectively reduced the viability of CL-6 and HuCCT-1 CCA cell lines, with selectivity indices (SI) of 3.53 and 2.61, respectively, outperforming free ATD and 5-fluorouracil (5-FU). They suppressed CL-6 cell migration and invasion by up to 90% within 12 h and induced apoptosis in 83% of cells through caspase-3/7 activation. Micronucleus assays showed lower mutagenic potential than the positive control. In vivo, ATD-PLGA NPs dose-dependently inhibited tumor growth and prolonged survival in CCA-xenografted nude mice; the high-dose regimen matched or exceeded the efficacy of 5-FU. Gene expression analysis revealed significant downregulation of pro-tumorigenic factors (VEGF, MMP-9, TGF-β, TNF-α, COX-2, PGE_2_, and IL-6) and upregulation of the anti-inflammatory cytokine IL-10. Collectively, these results indicate that ATD-PLGA NPs are a promising nanotherapeutic platform for targeted CCA treatment, offering improved anticancer potency, selectivity, and safety compared to conventional therapies.

## 1. Introduction

Cholangiocarcinoma (CCA) is the second most common primary hepatic malignancy, representing roughly 15% of liver cancers worldwide [1]. Its highly aggressive biology, late clinical presentation, and intrinsic resistance to conventional chemotherapy limit the five-year overall survival rate to <10% [2,3]. Curative surgical resection is rarely feasible because most patients present with locally advanced or metastatic disease. At the same time, first-line gemcitabine—or 5-fluorouracil-based regimens—confers only modest survival benefits and substantial toxicity [4]. These shortcomings underscore the urgent need for novel, tumor-selective, and better-tolerated therapies.

Traditional herbal medicines have re-emerged as valuable reservoirs of anticancer leads due to their multi-target actions and favorable safety profiles. *Atractylodes lancea* (Thunb.) DC—widely used in Chinese, Japanese, and Thai medicine—has long been prescribed for gastrointestinal and inflammatory ailments [5,6]. Its principal constituent, atractylodin, exhibits anti-inflammatory, antioxidative, and anticancer properties in diverse experimental systems [7]. Unfortunately, atractylodin’s poor aqueous solubility, rapid metabolism, and low bioavailability have hindered its clinical translation [8].

Nanotechnology offers powerful solutions to these limitations. Poly (lactic co-glycolic acid) (PLGA) nanoparticles—approved by the U.S. FDA—combine biodegradability, biocompatibility, and controlled drug release, thereby improving systemic exposure and tumor accumulation of payloads [9]. Encapsulating plant-derived molecules, such as atractylodin, in PLGA nanocarriers could enhance pharmacokinetics, minimize off-target toxicity, and permit rational modification for tumor-microenvironment targeting. This approach is particularly attractive for CCA, whose desmoplastic, immune-evasive microenvironment is sustained by chronic inflammation and tumor-promoting cytokines [10]. Here, we report the development, physicochemical characterization, and preclinical evaluation of atractylodin-loaded PLGA nanoparticles (ATD-PLGA NPs) as a targeted therapeutic platform for CCA in both cell-based and xenograft models. This study provides a proof-of-concept for using nanoencapsulation to overcome pharmacological limitations of traditional herbal compounds. It establishes ATD-PLGA NPs as a promising targeted therapy for a highly aggressive cancer with limited treatment options.

## 2. Materials and Methods

### 2.1. Chemicals and Reagents

Atractylodin (99% purity) was purchased from Shanghai Run-Biotech Co., Ltd. (Shanghai, China), and the control drug, 5-Fluorouracil (5-FU), was purchased from Wako Pure Chemical Industries Ltd. (Osaka, Japan). Both compounds were freshly prepared by dissolving in 50% ethanol (Labscan, Bangkok, Thailand). PLGA (poly lactic-co-glycolic acid) (50:50: MW 12,000, Resomer 504) and MTT reagent [3-(4,5-dimethylthiazol-2-yl)-2,5-diphenyltetrazolium bromide] were purchased from Sigma-Aldrich Inc. (St. Louis, MO, USA), and DMSO used in the MTT assay was obtained from Amresco LLC (Solon, OH, USA). The cell culture medium RPMI and DMEM, fetal bovine serum, and the antibiotics-antimycotics were purchased from Gibco BRL Life Technologies (Grand Island, NY, USA).

### 2.2. Cell Lines and Culture

The cholangiocarcinoma cell lines CL-6 and HuCCT-1 were cultured in RPMI 1640 medium supplemented with 10% heated fetal bovine serum and 100 IU/mL of antibiotic-antimycotic (anti-anti). Normal human embryo fibroblast cells (OUMS-36T-1F) were used, cultured in complete DMEM medium supplemented with 10% fetal bovine serum and 100 IU/mL antibiotic-antimycotic (anti-anti). All cells were maintained at 37 °C in a 5% CO_2_ atmosphere with 95% humidity.

### 2.3. Preparation of ATD-PLGA NPs

Poloxamer 407 is biocompatible and FDA-approved for pharmaceutical applications, making it suitable for translational research. Additionally, its amphiphilic nature facilitates particle size control and homogeneous distribution during nanoprecipitation. During the preparation of ATD-PLGA nanoparticles (NPs), a slight modification was incorporated into the nanoprecipitation method. The organic solution was formed by dissolving 2 mg of ATD and 50 mg of PLGA in 2 mL of acetone. The solution was carefully added to a 15 mL sample of deionized water containing 1% *w*/*v* poloxamer 407, which was stirred at a speed of 550 rpm. An infusion pump was utilized to maintain a drop rate of 10 mL per hour. To obtain a suspension of ATD-PLGA nanoparticles, the mixture was subjected to centrifugation at 13,000 rpm and 4 °C for 10 min. Ultimately, the liquid portion was discarded, while the solid part was recombined with purified water and stored in a refrigerator at 4 °C for later analysis.

### 2.4. Characterization of ATD-PLGA NPs

All nanoparticle formulations were characterized for average particle size, polydispersity index, and zeta potential using dynamic light scattering (DLS) with a Zetasizer (Malvern Instruments Ltd., Worcestershire, UK). To measure the nanoparticle diameter, surface charge, and polydispersity index, 1 mL of the suspension was added to a microcentrifuge tube and centrifuged at 13,000 rpm and 4 °C for 10 min. The supernatant was then discarded, and the pellet was resuspended in 1 mL of deionized water. The results were based on the average of three experiments. For morphology analysis, Transmission Electron Microscopy (TEM) was employed. A few microliters of ATD-PLGA nanoparticle suspension were placed on a carbon-coated 300-mesh copper grid and allowed to dry. The analysis was conducted using the digital Micrograph and Soft Imaging Viewer software, and the image was captured and analyzed.

### 2.5. Determination of Percent EE and LE

ATD-nanoparticle suspensions (1 mL) were added into microcentrifuge tubes and centrifuged at 13,000 rpm for 10 min at 4 °C. The supernatant was discarded, and 1 mL of DMSO was added to dissolve the pellet. The mixture was sonicated until it became a solution. The concentration of ATD was determined at a maximum wavelength of 340 nm by using a UV spectrophotometer. The percentages of EE (%EE) and LE (%LE) were calculated as follows:Encapsulation efficiency (%EE) = ((Amount of drug encapsulated in NPs)/(Total drug)) × 100Loading efficiency (%LE) = ((Amount of drug encapsulated in NPs)/(Total Polymer)) × 100

### 2.6. In Vitro Release Study of ATD-PLGA NPs

The determination of ATD-PLGA NPs was conducted under sink conditions. The drug-loaded nanoparticle, consisting of 300 μL (equivalent to 2 mg of loaded ATD), underwent centrifugation followed by suspension in 300 μL of phosphate buffer at pH 7.4. To simulate peristaltic conditions in vitro, the samples were placed in a Biosan medical-biological Research & Technologies device (Riga, Latvia). The shaker was set at a speed of 150 rpm and a temperature of 37 °C for a duration of 7 days. The sample was evaluated for the release of ATD by withdrawing 100 µL aliquots at specific time intervals (1, 3, and 6 h and 1, 2, 3, 4, 5, 6, and 7 days). UV spectrophotometry at a wavelength of 340 nm was used to estimate the amount of ATD released.Cumulative release of ATD (%) = ((DL − DR)/DL) × 100
where DL is the amount of drug loaded in NPs, and DR is the amount of drug that remains in NPs.

### 2.7. Cytotoxic Activity

CL-6, HuCCT-1, and OUMS-36T-1F cells were plated in 96-well culture plates (10,000 cells/well). After 24 h of incubation, the cells were incubated with ATD and ATD-PLGA NPs at concentrations ranging from 250 to 1.95 µM (via serial dilution) at 37 °C for 48 h. 5-Fluorouracil (concentration range: 3.90–500.00 µM) was used as a positive control drug. Cells were washed three times with phosphate-buffered saline (PBS), incubated with MTT solution (20 µL, 5 mg/mL) at 37 °C for 4 h, and then lysed with DMSO. Absorbance (optical density: OD) was read with a plate reader machine (Varioscan Flash, Thermo Fisher Scientific, Waltham, MA, USA) at 570 nm. The percentage of inhibition of the cytotoxic activity was calculated as follows:% Inhibition = (Absorbance control × Absorbance test) × 100

Absorbance control

The IC_50_ values were calculated using CALCUSYN^TM^ software version 2.0 (Biosoft, Cambridge, UK).

### 2.8. Cell Migration Assay

The xCELLigence RTCA DP Instrument (ACEA Biosciences, San Diego, CA, USA), also known as the real-time cell analyzer dual purpose, was used to explore a cell migration assay. The CIM plate was assembled and equilibrated at 37 °C for 1 h, after which the plate background was measured. The plate was first supplemented with CL-6 cells at a density of 50,000 cells per well. Subsequently, ATD-PLGA NPs were introduced in a serial dilution, with concentrations varying from 250 to 1.95 µM. The plate was placed in the xCELLigence RTCA DP system and incubated at 37 °C for 24 h, during which cell migration was continuously monitored. The measurement was ceased after 24 h of incubation, and the cell index curves were examined to ascertain cell migration activity.

### 2.9. Cell Invasion Assay

The two-day assay protocol was developed for continuous monitoring of cell invasion on the xCELLigence RTCA DP system for 24 h. The cell invasion/migration (CIM) plate was coated with Matrigel (BD Matrigel Basement Membrane Matrix, BD Biosciences, Franklin Lakes, NJ, USA) and placed in a 37 °C incubator for 4 h. The xCELLigence RTCA DP system enhances accuracy by providing label-free, real-time monitoring of CCA cell invasion through continuous impedance-based measurements. Unlike traditional endpoint assays that capture only a single time point, this system tracks dynamic cellular responses to ATD-PLGA NPs over time, enabling precise assessment of the onset, duration, and magnitude of anti-invasive effects with improved reproducibility and sensitivity. After 4 h of incubation, the CIM plate was assembled and equilibrated at 37 °C for 1 h, and the plate background was measured. CL-6 cells were added to the plate (50,000 cells/well), and then ATD-PLGA NPs were added at concentrations ranging from 250 to 1.95 µM (via serial dilution). The plate was incubated at 37 °C with continuous monitoring of cell invasion using the xCELLigence RTCA DP system for 24 h. After 24 h of incubation, the plate was stopped, and the cell index curves were analyzed to determine cell invasion activity.

### 2.10. Apoptosis Activity

CL-6 cells were plated in 24-well plates at an initial density of 100,000 cells/well and then treated with ATD (220 μM) and ATD-PLGA NPs at concentrations of 75 and 150 uM and maintained at 37 °C in a 5% CO_2_ atmosphere with 95% humidity for 48 h. 5-Fluorouracil (concentration of IC_50_) was used as a positive control drug. After treatment, cells were labeled with 5 μM CellEvent^TM^ caspase-3/7 Green Detection Reagent (Molecular Probes Invitrogen, Carlsbad, CA, USA) in complete medium for 30 min at 37 °C in the dark. Stained cells were observed using an InCell Analyzer 6000 cell imaging system (GE Healthcare Life Sciences, Piscataway, NJ, USA).

### 2.11. Mutagenic Assay

The CL-6, HuCCT-1, and OUMS-36T-1F cell lines were seeded into 6-well plates at a density of 2 × 10^5^ cells/mL and incubated at 37 °C for 24 h under 5% CO_2_. The supernatant was removed, and fresh medium was added to each well. The CL-6 and HuCCT-1 cells were treated with different concentrations of ATD and ATD-PLGA NPs at the IC_50_. The OUMS-36T-1F cell was treated with an IC10 dose of ATD and ATD-PLGA NPs. Mitomycin C (MMC) was used as a positive control at the concentration of 0.5 μg/mL for OUMS-36T-1F and 1 μg/mL for CL-6 and HuCCT-1 cells, respectively. All cells were incubated at 37 °C under 5% CO_2_ for 4 h, and the supernatant (90 µL) was removed. Cytochalasin B (90 µL of 3 µg/mL) was added into each well, carefully swirled, and further incubated for an additional 24 h. Cells were washed twice with PBS and trypsinized. Cold Hank’s balanced salt solution (HBSS: 2 mL) (Corning, Corning, NY, USA) was added to each tube and centrifuged at 2360× *g* (Kubota 5922 laboratory centrifuge, Tokyo, Japan) for 2 min. Cell supernatant was discarded, and HBSS (300 µL) was added to each well and thoroughly mixed. Cells were prepared as monolayers on glass slides using Cytospin equipment (Shandon, UK). Slides were left to dry at room temperature (25 °C), then fixed in cold methanol for 30 min, and subsequently left to dry. Cells were stained with 10% Giemsa stain for 20 min. The excess Giemsa stain was carefully washed with GURR buffer (Gibco Invitrogen, Carlsbad, Germany) and left to dry overnight. Thereafter, permount^®^ SP15-500 toluene solution was dropped into each slide and covered with a coverslip. The micronucleus (MN) frequency was analyzed by scoring 1000 binucleated (BNC) cells per treatment under a light microscope (40×) for the presence of micronuclei. The cytotoxic effect of these cells was determined and expressed as the cytokinesis-block proliferation index (CBPI) value. This CBPI value was considered the cell kinetic, or average number of cell divisions, which was determined by scoring 500 cells and classifying them according to the number of nuclei. The CBPI value was calculated using the following formula:CBPI = [(number of mononucleated cell) + (number of BNC cell × 2) + (number of multicleated cell × 3)]/total number of cells scored

### 2.12. Animal Model

The animal study was approved by the Institutional Animal Care and Use Committee, Thammasat University, under Ethics Approval No. 002/2021. To induce an animal model for CCA, the human CCA cell line CL-6 was used for tumor xenografting in nude mice. After being cultivated in RPMI 1640 media, CL-6 cells were scraped from the culture flask using a cell scraper. After gathering all the cells into a 15 mL conical tube, they were centrifuged for 5 min at 25 °C at 100× *g*. Following removal, the cell supernatant was resuspended in 3 mL of medium. A hemocytometer chamber was used to count the number of cells. After the injection site was cleaned, cells for injection (1,000,000 cells/200 μL in complete media) were prepared and subcutaneously injected into the right upper flanks of nude mice. Before the trial, body weight and tumor size were monitored every two days, and the mice were examined daily.

Based on the MTD of ATD [11], mice were randomized into three dose groups: 100, 50, and 25 mg/kg body weight of ATD-PLGA NPs. 5-FU (40 mg/kg body weight) and a control vehicle were administered to the control groups. Each group consisted of six mice, paired and matched based on the size of the tumor (when tumor nodules reached a volume of roughly 50–100 mm^3^). For thirty days, the animals were fed all test chemicals every day by intragastric gavage. The study protocol was approved by the Ethics Committee for Animal Research of Thammasat University, Thailand (Number 002/2021).

### 2.13. mRNA Expression by Quantitative Real-Time PCR

The tumors were washed with PBS and then immediately prepared for RNA extraction. The RNA underwent processing using the RNeasy Mini Kit (Qiagen, Hilden, Germany). To eliminate any trace of genomic DNA, the total RNA underwent treatment with the RQ1 RNase-Free DNase Kit (Promega, Madison, WI, USA). Real-time PCR was used to quantify the gene expression relatively. To amplify the housekeeping gene control GAPDH and eight specific target genes of interest, three sets of primers were used in all reactions. These sets of primers were used in Table 1.

All gene copy numbers will be determined by SYBR Green I real-time PCR (iCycler™, Bio-Rad, Hercules, CA, USA) using the default thermocycler program. All primers will be used as Table 1. The sequences of these PCR products will be obtained by direct sequencing. Each amplicon will be cloned into the PGEM-t Vector (Promega, Madison, WI, USA) to create standard curves for the target cDNA. The mRNA levels will be reported as ratios of the copy numbers of target cDNAs to GAPDH cDNA.

### 2.14. Statistical Analysis

All quantitative variables are presented as median (range) values. Comparison of all quantitative variables between the groups given test materials or reference drugs was performed using the student t-test. The statistical significance level was set at α = 0.05 for all tests.

## 3. Results

### 3.1. Preparation of ATD Nanoparticles

The ATD-PLGA NPs produced by the nanoprecipitation method exhibited a narrow size distribution with homogeneous size and a negative zeta potential. The median (range) particle size of the characterized ATD-PLGA NPs was 229.8 (225.6–233.3) nm, with a PDI of 0.112 (0.076–0.114) and a zeta potential of −25.3 (−25.2–27.9) mV, compared to blank PLGA NPs with a corresponding particle size of 225.7 (220.1–224.3) nm, a PDI of 0.122 (0.118–0.137), and a zeta potential of −29.5 (−29.4–30.3) mV. A high percentage of EE and LE was obtained for ATD-PLGA NPs, with values of 83% (82–85%) and 3.5% (3.3–3.6%), respectively. The characterized ATD-PLGA NPs values were statistically significant compared to the blank PLGA NPs (*p* < 0.05). The shape and morphology of ATD-PLGA were evaluated using TEM (Figure 1). The biphasic release rate profile of ATD-PLGA NPs exhibited an initial rapid release of 80% within the first 24 h, which was followed by a sustained release of 90% over 48 h and 92% throughout 7 days.

### 3.2. Cell Cytotoxicity

The cytotoxic effects of 5-fluorouracil (5-FU), ATD, and ATD-loaded PLGA nanoparticles (ATD-PLGA NPs) were assessed in two CCA cell lines (CL-6 and HuCCT-1) and one normal fibroblast cell line (OUMS-36T-1F) using the MTT assay. The median IC_50_ values and selectivity index (SI) values are summarized in Table 2. In the CL-6 cell line, ATD-PLGA NPs exhibited the most remarkable cytotoxicity, with an IC_50_ of 150.63 µM (range: 155.85–178.35 µM), followed by ATD at 220.74 µM (195.42–231.68 µM), and 5-FU at 525.39 µM (508.44–598.16 µM). Corresponding SI values were 3.53 for ATD-PLGA NPs, 2.00 for ATD, and 1.95 for 5-FU.

In HuCCT-1 cells, a similar trend was observed. ATD-PLGA NPs showed enhanced potency with an IC_50_ of 203.85 µM (198.41–247.88 µM), compared to ATD (280.46 µM; 269.82–302.39 µM) and 5-FU (675.07 µM; 635.89–699.25 µM). The SI values were 2.61, 1.57, and 1.52, respectively (Table 2).

For the normal cell line OUMS-36T-1F, all treatments displayed lower cytotoxicity, with IC_50_ values of 1025.42 µM (989.48–1090.93 µM) for 5-FU, 441.55 µM (420.65–465.77 µM) for ATD, and 532.48 µM (476.31–568.04 µM) for ATD-PLGA NPs. The SI for all agents in this cell line was normalized to 1.00 (Table 2).

These results indicate that ATD-PLGA NPs exhibit enhanced selective cytotoxicity against CCA cells compared to both free ATD and 5-FU and greater selectivity in CL-6 cells, suggesting the potential of nanoparticle-based delivery in targeted cancer therapy.

### 3.3. Inhibitory Effect on Cell Migration and Cell Invasion

Figure 2 shows the time- and dose-dependent directional kinetics of migration/invasion of the CL-6 cells in real time using the xCELLigence RTCA. With this assay, the quantity of invasive or migratory cells may be tracked in real-time and displayed as the cell index. The cell index profiles revealed the rapid onset of ATD-PLGA NPs-inhibited CL-6 cell migration and cell invasion continuously (Figure 2A,B). CL-6 cells were significantly (*p* < 0.05) 4 times more migratory and invasive than control cells at the 12 h time point following treatment with ATD-PLGA NPs. The kinetics pattern of cell migration and invasion after exposure to the standard drug 5-FU differed from that of ATD-PLGA NPs, which inhibited CL-6 cells after 16 and 9 h, respectively (Figure 2A,B).

The results were expressed as a percentage inhibition by comparing them to the control group. The inhibition of cell migration values from ATD-PLGA NPs (75 and 150 µM), ATD, and 5-FU-treated groups at 12 h were 90%, 80%, 25%, and 0%, respectively. The inhibition of cell invasion values from ATD-PLGA NPs (75 and 150 µM), ATD, and 5-FU-treated groups at 12 h were 82%, 23%, 28%, and 0%, respectively. Interestingly, the inhibitory effect of ATD-PLGA NPs was decreased after 24 h of incubation (Figure 2A,B).

### 3.4. Inducing Effect on Cell Apoptosis

Evaluation of cell apoptosis was performed by fluorescence staining of caspase 3 and caspase 7 activities, which are involved in the cell apoptotic pathway. The green signal was observed at all concentrations of ATD and ATD-PLGA NPs. The number of cells with green fluorescence that underwent apoptosis was counted and compared to the total number of cells, which was approximately 1000 (Figure 3). ATD (220 μM) and ATD-PLGA NPs at concentrations of 75 and 150 μM, and 5-FU, produced 48%, 61%, 83%, and 40% inducing effects on CL-6 cell apoptosis, respectively (Figure 4).

### 3.5. Mutagenic Activity

The genotoxicity of ATD and ATD-PLGA NPs was investigated using the micronucleus assay. The characteristics of a normal cell and a micronucleus are shown in Figure 5. Following exposure to MMC for 24 h, the number of micronuclei was increased in the CL-6 cell line compared to untreated cells. To assess cell proliferation, the cytokinesis block proliferation index (CBPI) was used to determine the ability of the treated cells to proliferate during the assay. The CBPI values of all cell lines were greater than one, suggesting complete cell division. After treatment with test materials, the number of micronuclei was observed to be lower in the ATD-PLGA NPs and ATD-treated groups compared with the MMC-induced control.

### 3.6. Anti-Cholangiocarcinoma Activity

The median (range with 95%CI) values of body weight at the end of treatment (on day 31) for the 5-FU-treated and untreated control groups were 34 (31–38) and 46 (42–48) g, respectively. The body weights of the ATD-PLGA NPs-treated groups at high-, medium-, and low-dose levels were 26 (25–30), 33 (30–37), and 45 (40–48) g, respectively.

Treatment with ATD-PLGA NPs resulted in a marked reduction in tumor size across all dose levels in a dose-dependent manner (Figure 6). Mice receiving the high-dose ATD-PLGA NPs (100 μM) exhibited the most pronounced tumor shrinkage, with tumor morphology showing significant degradation compared to the untreated group. The medium- and low-dose groups also demonstrated tumor suppression, albeit to a lesser extent. In terms of survival outcomes, mice treated with ATD-PLGA NPs displayed extended survival times compared to the untreated controls (Figure 6). Notably, the high-dose group achieved the longest survival time among all treatment groups. The therapeutic performance of ATD-PLGA NPs at high dose was comparable to, or exceeded, that observed with 5-FU, indicating potent anticancer efficacy. These results support the potential of ATD-PLGA nanoparticles as a promising therapeutic agent for treating cholangiocarcinoma.

### 3.7. mRNA Expression

The mRNA expression levels of eight genes (VEGF, MMP9, TGF-β, TNF-α, COX-2, PGE2, IL-6, and IL-10) were evaluated in CCA-xenografted nude mice following treatment with ATD-PLGA nanoparticles. The expression levels of pro-inflammatory and tumor-related genes, including VEGF, MMP9, TGF-β, TNF-α, COX-2, PGE2, and IL-6, were significantly downregulated in a dose-dependent manner following treatment with ATD-PLGA NPs compared to the untreated group. The high-dose ATD-PLGA NPs group exhibited the most pronounced suppression, with expression levels comparable to or greater than those observed in the 5-FU-treated group (Figure 7).

In contrast, the anti-inflammatory cytokine IL-10 was upregulated in ATD-PLGA NP-treated groups, particularly in the high-dose group, indicating potential anti-inflammatory activity of the formulation (Figure 7). These findings suggest that ATD-PLGA NPs exhibit dose-dependent modulation of gene expression associated with tumor progression and inflammation, comparable to that of the standard chemotherapeutic agent 5-FU.

## 4. Discussion

The present study successfully formulated and characterized atractylodin-loaded poly(lactic-co-glycolic acid) nanoparticles (ATD-PLGA NPs), establishing a versatile nanomedicine platform that addresses the poor aqueous solubility of ATD and reprograms its pharmacological profile against cholangiocarcinoma (CCA) [12,13]. Utilizing a streamlined nanoprecipitation technique, the ATD-PLGA NPs exhibited favorable physicochemical properties, including a uniform particle size of approximately 230 nm, a low polydispersity index (PDI < 0.12), and a high encapsulation efficiency (~83%). These parameters align well with the optimal criteria for systemic nanocarrier delivery, as the size is ideal for evading rapid renal clearance while avoiding hepatic filtration, thereby facilitating tumor penetration through the enhanced permeability and retention (EPR) effect [9,14]. The biphasic release pattern—characterized by an initial burst followed by sustained drug release—mirrors clinically approved PLGA systems, enabling rapid tumor priming and prolonged intratumoral drug exposure [15], which are crucial for maximizing antitumor efficacy and reducing dosing frequency [16]. This biphasic profile ensures early tumor exposure to atractylodin and maintains prolonged cytotoxic levels over time. This contrasts with single-phase or immediate-release formulations, which may lead to short-lived efficacy, rapid clearance, and reduced tumor retention.

Altogether, these favorable physicochemical attributes translated into significant biological advantages. In vitro, ATD-PLGA NPs demonstrated three- to five-fold lower IC_50_ values compared to free ATD and outperformed the conventional chemotherapeutic 5-fluorouracil (5-FU) in CCA cell lines CL-6 and HuCCT-1. Importantly, cytotoxicity towards normal fibroblast cells (OUMS-36T-1F) remained minimal, reflecting a favorable safety profile and enhanced selectivity. This improved therapeutic index is likely driven by enhanced cellular internalization, efficient endosomal escape, and sustained intracellular release of ATD [17]. Functional assays revealed that ATD-PLGA NPs rapidly and sustainably inhibited cell migration and invasion within the first 12 h, indicating early interference with metastatic processes—an effect likely mediated by improved retention of the drug in tumor cells [18]. Furthermore, elevated caspase-3/7 activity confirmed robust apoptosis induction, reinforcing ATD’s pro-apoptotic effects, as reported in other cancer models [19]. The nanoparticle formulation alters the distribution and metabolism of atractylodin by encapsulating it within PLGA, which protects the compound from rapid enzymatic degradation and first-pass metabolism. This results in prolonged systemic circulation and enhanced bioavailability. Additionally, the nanoscale size facilitates passive tumor targeting via the enhanced permeability and retention (EPR) effect, allowing preferential accumulation at tumor sites. In contrast, free atractylodin is rapidly metabolized and cleared, leading to lower therapeutic levels and reduced efficacy. The distinct kinetic pattern of ATD-PLGA NPs, showing rapid and sustained inhibition of CL-6 cell migration and invasion, reflects their enhanced cellular uptake and prolonged intracellular release of atractylodin. This enables early and continuous disruption of motility-related signaling pathways, unlike free atractylodin or 5-FU, which act more slowly or indirectly due to differences in uptake, stability, and target specificity.

The enhanced anticancer efficacy of ATD-PLGA NPs was recapitulated in vivo using a subcutaneous CCA xenograft model [20]. The nanoparticles induced dose-dependent tumor volume reduction and extended overall survival by over 40%, with the high-dose group demonstrating efficacy comparable to or exceeding that of 5-FU without associated systemic toxicity or weight loss. Molecular analyses supported these findings, showing significant downregulation of tumor-promoting genes, including VEGF, MMP-9, and IL-6, as well as upregulation of the anti-inflammatory cytokine IL-10. This suggests that ATD-PLGA NPs exert multifaceted antitumor effects by simultaneously inhibiting angiogenesis, tumor invasiveness, and inflammation [21,22]. Treatment with ATD-PLGA NPs leads to stronger dose-dependent downregulation of pro-tumorigenic genes and upregulation of IL-10 compared to 5-FU due to the sustained release, enhanced cellular uptake, and multi-targeted action of atractylodin. Unlike 5-FU, which primarily targets DNA synthesis, ATD-PLGA NPs modulate inflammatory and metastatic signaling pathways more effectively, contributing to broader and more potent gene regulatory effects. Moreover, micronucleus assays revealed lower genotoxicity than mitomycin C and preserved mitotic activity, further attesting to the biocompatibility and safety of the nanoparticle formulation [23]. The reduced number of micronuclei in ATD-PLGA NP–treated cells compared to the MMC control indicates a lower genotoxic potential, suggesting that the nanoparticles do not significantly damage DNA or interfere with chromosomal integrity. This implies greater long-term safety for therapeutic applications, as it reduces the risk of mutation-related side effects or secondary malignancies during prolonged use.

While these findings are promising, several limitations warrant consideration. The subcutaneous xenograft model does not fully replicate the complex, hypovascular, and fibrotic microenvironment of intrahepatic CCA, emphasizing the need for future studies using orthotopic or patient-derived xenograft (PDX) models to mimic clinical pathology better [24]. The relatively short observation period of up to 40 days may not capture long-term therapeutic effects or late-onset toxicities, underscoring the importance of extended follow-up studies. Additionally, the lack of detailed pharmacokinetic and biodistribution data limits our understanding of in vivo nanoparticle behavior; therefore, comprehensive studies using labeled nanoparticles, combined with physiologically based pharmacokinetic (PBPK) modelling, are essential for dose optimization and translation to humans [25]. Furthermore, this study focused exclusively on ATD as a monotherapy; given the multifactorial nature of CCA, future work should explore combination therapies involving ATD-PLGA NPs with standard chemotherapeutics such as gemcitabine or emerging immunotherapies to evaluate potential synergistic effects. Lastly, while the nanoprecipitation technique yielded high-quality nanoparticles at laboratory scale, the feasibility of scalable, GMP-compliant manufacturing processes must be addressed to ensure clinical applicability.

## 5. Conclusions

Encapsulating ATD into PLGA nanoparticles significantly enhanced its anticancer efficacy and selectivity against CCA cells by improving ATD’s solubility, stability, and sustained release profile, which collectively increased its intracellular retention and tumor-specific accumulation. Unlike free ATD and 5-FU, the nanoparticle formulation enabled efficient cellular uptake, prolonged drug exposure, and targeted modulation of cancer-related pathways—resulting in stronger cytotoxicity toward CCA cells with reduced toxicity to normal cells. The key innovative advantage of using atractylodin-loaded PLGA nanoparticles lies in their ability to combine the multi-targeted anticancer properties of a natural compound with the precision and sustained delivery of a clinically approved nanoparticle system. Unlike existing CCA treatments or other nanoparticle platforms, this formulation offers enhanced selectivity, reduced toxicity, and immunomodulatory effects, positioning it as a safer and more effective alternative for targeted cholangiocarcinoma therapy. To fully realize this potential, future investigations should prioritize the optimization of in vivo pharmacokinetics, evaluation in more clinically relevant tumor models, exploration of rational combination regimens, and advancement toward scalable production. Collectively, this nanomedicine strategy offers a compelling path forward to enrich the therapeutic arsenal against this challenging and aggressive cancer.

## Figures and Tables

**Figure 1 polymers-17-02151-f001:**
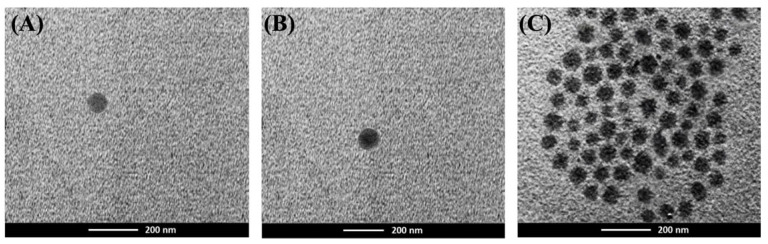
Transmission Electron Microscopy (TEM) images of (**A**) ATD, (**B**) ATD-PLGA NPs, and (**C**) aggregation of ATD-PLGA NPs.

**Figure 2 polymers-17-02151-f002:**
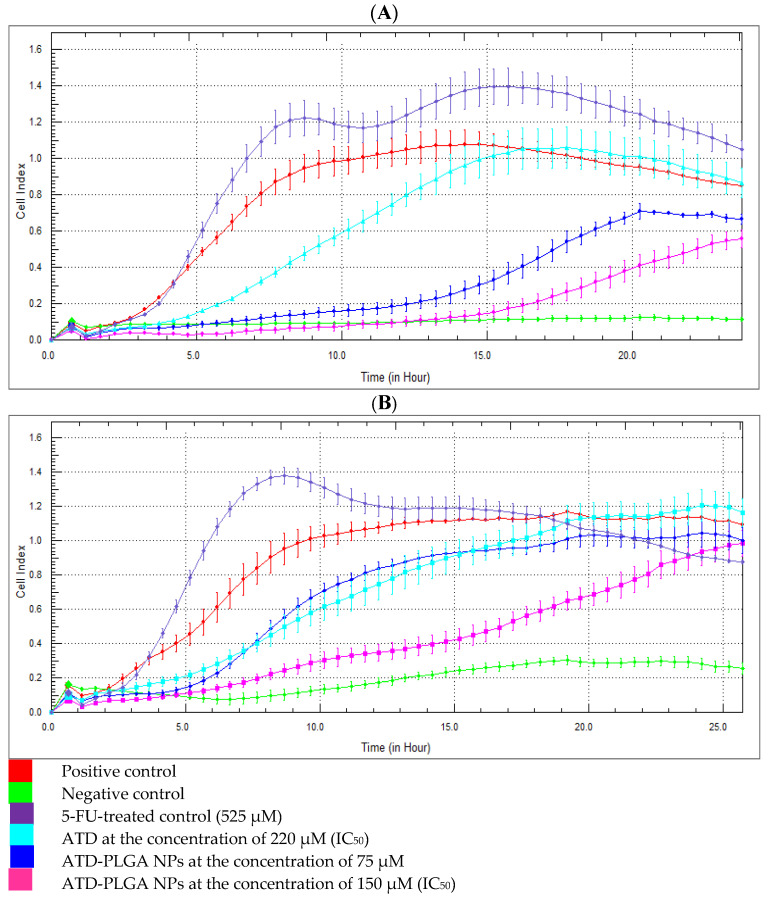
Time- and dose-dependent directional migration (**A**) and invasion (**B**) of CL-6 cells using xCELLigence Real-Time Cell Analyzer (RTCA) DP by treating with ATD (blue: 220 μM), ATD-PLGA NPs (dark blue: 75 μM; pink: 150 μM) compared with positive (red) and negative (green) and 5-FU-treated (purple) control groups (*p* < 0.001).

**Figure 3 polymers-17-02151-f003:**
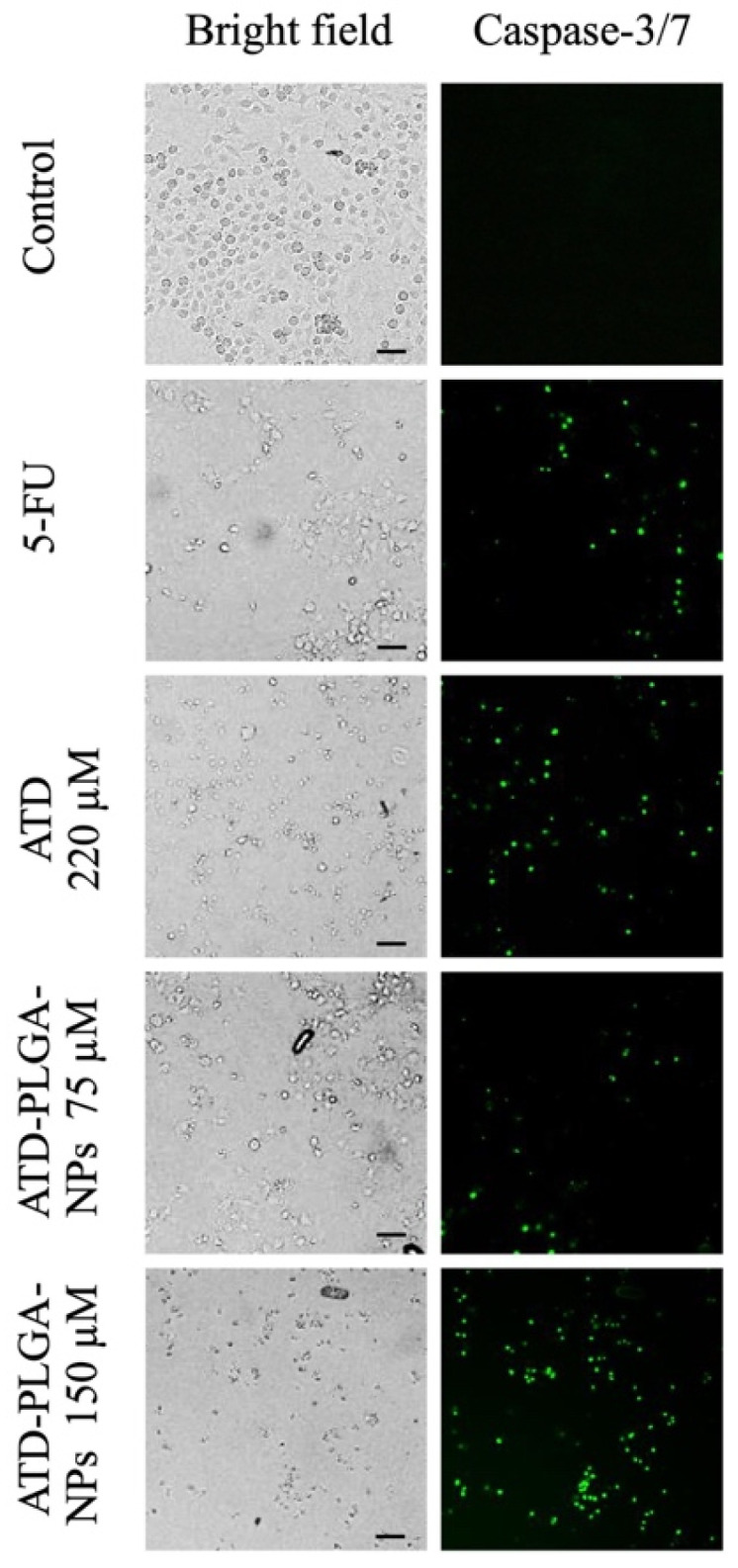
Morphological characteristics under a fluorescent microscope of CL-6 cells apoptosis by using CellEvent^TM^ Caspase-3/7 Green Detection Reagent. Left panels are bright fields, and green fluorescence channels are shown in right panels (scale bar  =  20 µM).

**Figure 4 polymers-17-02151-f004:**
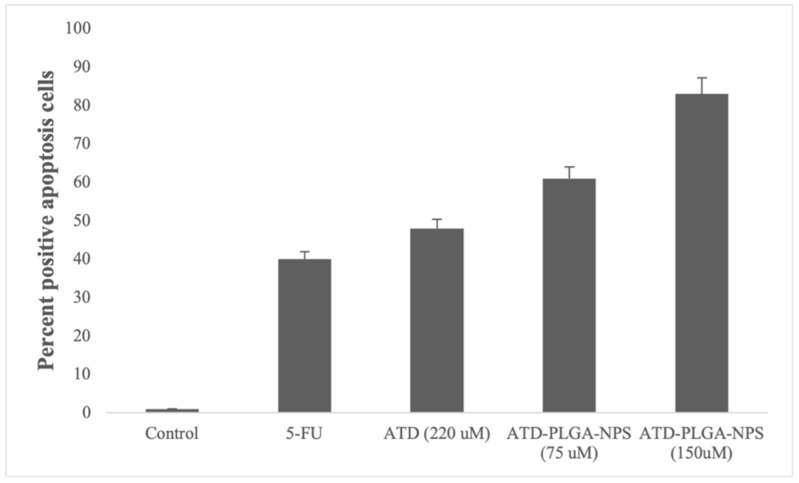
The apoptosis event percentage is calculated from the Caspase-3/7 activities by using CellEvent™ caspase-3/7 compared with the control in 5-FU, ATD (220 uM), and ATD-PLGA NPs (75 and 150 uM)-treated cells, respectively.

**Figure 5 polymers-17-02151-f005:**
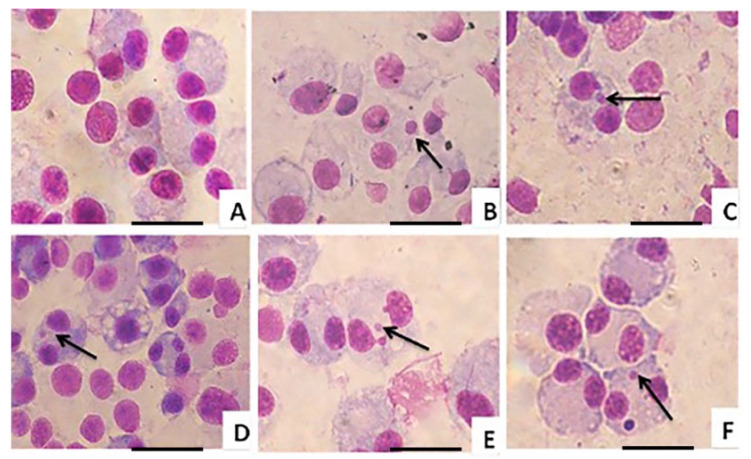
Morphology of the micronuclei of the control CL-6 cells (**A**), 1 µg/mL MMC-treated cells (**B**), 5-FU-treated cells (**C**), ATD-treated cells (**D**), ATD-PLGA NPs (75 μM)-treated cells (**E**), and ATD-PLGA NPs 1(150 μM)-treated cells (**F**) (scale bar  =  20 µM).

**Figure 6 polymers-17-02151-f006:**
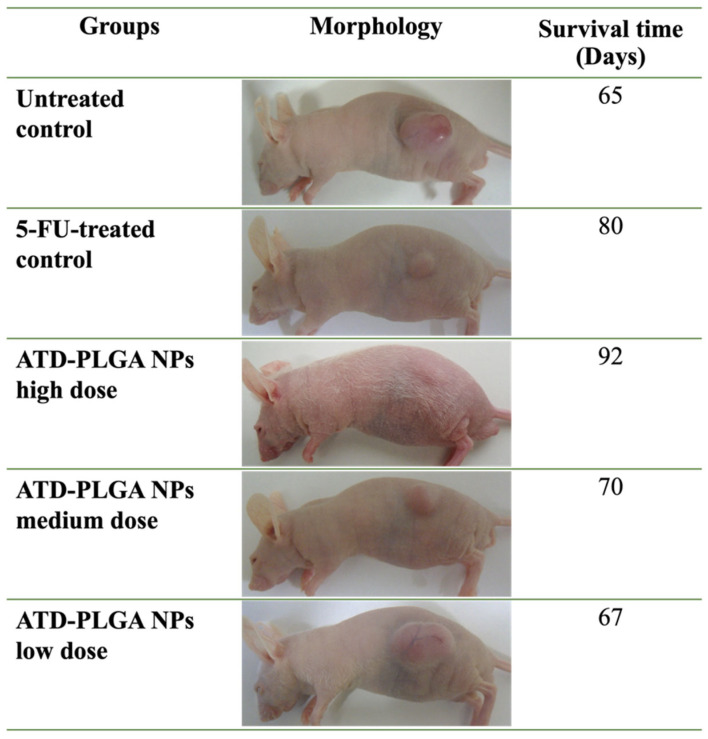
Representative tumor morphology and survival time from CCA-xenografted nude mice following treatment with ATD-PLGA NPs at high-, medium-, and low-dose levels (100, 50, and 25 mg/kg body weight, respectively), 5-fluorouracil (reference control: 40 mg/kg body weight), and untreated control.

**Figure 7 polymers-17-02151-f007:**
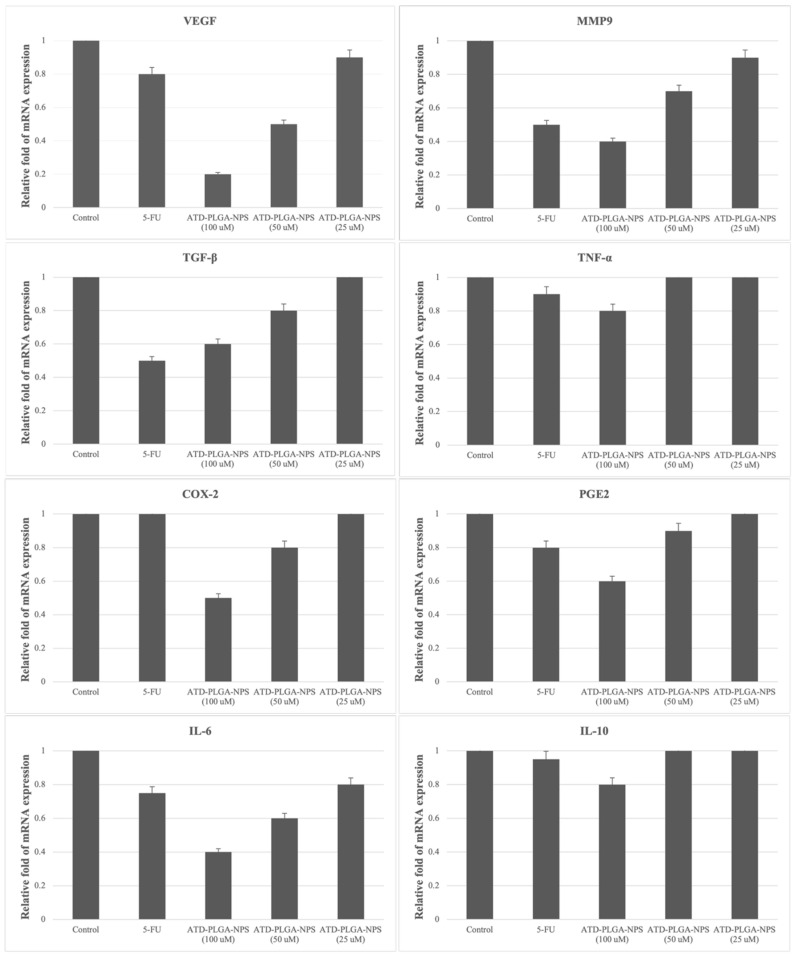
The levels of gene expression in CCA-xenografted nude mice following treatment with ATD-PLGA NPs at high-, medium-, and low-dose levels (100, 50, and 25 mg/kg body weight, respectively), 5-fluorouracil (reference control: 40 mg/kg body weight), and untreated control.

**Table 1 polymers-17-02151-t001:** Represent a set of eight primers for mRNA expression.

Gene	Forward Primer (5′–3′)	Reverse Primer (5′–3′)
VEGF	ACACATTGTTGGAAGAAGCAGCCC	AGGAAGGTCAACCACTCACACACA
MMP9	TCGAAGGCGACCTCAAGTG	TTCGGTGTAGCTTTGGATCCA
TGF-β	GGTTCATGTCATGGATGGTGC	TGACGTCACTGGAGTTGTACGG
TNF-α	TGAGGGATCTGTGGATGCTTCGT	AAACCCACAGTGCTTGACACAGAA
COX-2	TGAGGGATCTGTGGATGCTTCGT	AAACCCACAGTGCTTGACACAGAA
PGE_2_	CTTCCTTTTCCTGGGCTTCG	GAAGACCAGGAAGTGCATCCA
IL-6	CCAGCTATGAACTCCTTCTC	GCTTGTTCCTCACATCTCTC
IL-10	GCCTTATCGGAAATGATCCA	TCTCACCCAGGGAATTCCAAA
GAPDH	GGCCCACATGGCCTCCAAGG	GGCAGGGACTCCCCAGCAGT

**Table 2 polymers-17-02151-t002:** Summary of the IC50 and SI values for ATD and ATD-PLGA NPs by using MTT assay. Data presented as median (range) value.

Cell Type	Potency/ Selectivity	5-FU	ATD	ATD-PLGA NPs
CL-6	IC_50_ (µM)	525.39 (508.44–598.16)	220.74 (195.42–231.68)	150.63 (155.85–178.35)
	SI	1.95	2.00	3.53
HuCCT-1	IC_50_ (µM)	675.07 (635.89–699.25)	280.46 (269.82–302.39)	203.85 (198.41–247.88)
	SI	1.52	1.57	2.61
OUMS-36T-1F	IC_50_ (µM)	1025.42 (989.48–1090.93)	441.55 (420.65–465.77)	532.48 (476.31–568.04)
	SI	1	1	1

## Data Availability

The data presented in this study are available from the corresponding author upon request.

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
