# Peer review of "Enhanced Anticancer Activity of Atractylodin-Loaded Poly(lactic-co-glycolic Acid) Nanoparticles Against Cholangiocarcinoma"

_polymers, 2025, doi:10.3390/polym17152151_

Round 1

Reviewer 1 Report

Comments and Suggestions for Authors

The submitted paper presents a promising nanotherapeutic approach for cholangiocarcinoma by encapsulating atractylodin, a bioactive compound from Atractylodes lancea, into PLGA nanoparticles. The nanoparticles demonstrated favorable physicochemical properties and sustained drug release, leading to enhanced anticancer effects both in vitro and in vivo compared to free atractylodin and the conventional drug 5-FU. The treatment showed high selectivity, strong inhibition of tumor growth, induction of apoptosis, and favorable modulation of tumor-related gene expression, with lower mutagenic potential. In my opinion, the paper has been fairly organized and can be accepted for publication after a revision. My comments are as follow:
1)    The key innovative advantage of using atractylodin-loaded PLGA nanoparticles over existing cholangiocarcinoma treatments or other nanoparticle-based drug delivery systems should be explained.
2)    The abstract part has been made of two paragraphs. It should be one paragraph. 
3)    The resolution of Figure 1 should be improved. 
4)    It should be elaborated how the nanoparticle formulation affects the distribution and metabolism of atractylodin compared to its free form.
5)    In materials and method section, it should be explained why poloxamer 407 was chosen as the stabilizer during the preparation of ATD-PLGA nanoparticles.
6)    In page 4, please explain how the use of the xCELLigence RTCA DP system enhances the accuracy or real-time monitoring of ATD-PLGA NPs effect on CCA cell invasion compared to traditional endpoint assays.
7)    In page 6, how does the biphasic release profile of ATD-PLGA nanoparticles contribute to their therapeutic effectiveness compared to a single-phase or immediate-release formulation?
8)    In page 7, please explain how the distinct kinetic pattern of ATD-PLGA NPs in inhibiting CL-6 cell migration and invasion reflects differences in their mechanisms of action?
9)    In page 10, it should be explained how the reduced number of micronuclei in ATD-PLGA NP–treated cells compared to the MMC control indicates a lower genotoxic potential, and what are the implications for long-term safety in therapeutic applications.
10)    In page 11, please explain why treatment with ATD-PLGA NPs leads to a stronger dose-dependent downregulation of pro-tumorigenic genes and upregulation of IL-10 compared to conventional 5-FU.
11)    In page 13, it should be explained why encapsulating ATD into PLGA nanoparticles significantly enhanced its anticancer efficacy and selectivity against CCA cells compared to free ATD and 5-FU.
12)    The English of paper is acceptable.

Author Response

Reviewer 1:

The submitted paper presents a promising nanotherapeutic approach for cholangiocarcinoma by encapsulating atractylodin, a bioactive compound from Atractylodes lancea, into PLGA nanoparticles. The nanoparticles demonstrated favorable physicochemical properties and sustained drug release, leading to enhanced anticancer effects both in vitro and in vivo compared to free atractylodin and the conventional drug 5-FU. The treatment showed high selectivity, strong inhibition of tumor growth, induction of apoptosis, and favorable modulation of tumor-related gene expression, with lower mutagenic potential. In my opinion, the paper has been fairly organized and can be accepted for publication after a revision. My comments are as follow:

1)    The key innovative advantage of using atractylodin-loaded PLGA nanoparticles over existing cholangiocarcinoma treatments or other nanoparticle-based drug delivery systems should be explained.

Response: The key innovative advantage of using atractylodin-loaded PLGA nanoparticles lies in their ability to combine the multi-targeted anticancer properties of a natural compound with the precision and sustained delivery of a clinically approved nanoparticle system. Unlike existing CCA treatments or other nanoparticle platforms, this formulation offers enhanced selectivity, reduced toxicity, and immunomodulatory effects, positioning it as a safer and more effective alternative for targeted cholangiocarcinoma therapy. This information has been added to the manuscript.

2)    The abstract part has been made of two paragraphs. It should be one paragraph.

Response: This has been corrected.

3)    The resolution of Figure 1 should be improved.

Response: Figure 1 has been revised with high resolution.

4)    It should be elaborated how the nanoparticle formulation affects the distribution and metabolism of atractylodin compared to its free form.

Response: The nanoparticle formulation alters the distribution and metabolism of atractylodin by encapsulating it within PLGA, which protects the compound from rapid enzymatic degradation and first-pass metabolism. This results in prolonged systemic circulation and enhanced bioavailability. Additionally, the nanoscale size facilitates passive tumor targeting via the enhanced permeability and retention (EPR) effect, allowing preferential accumulation at tumor sites. In contrast, free atractylodin is rapidly metabolized and cleared, leading to lower therapeutic levels and reduced efficacy. This information jas been added to the manuscript.

5)    In materials and method section, it should be explained why poloxamer 407 was chosen as the stabilizer during the preparation of ATD-PLGA nanoparticles.

Response: Poloxamer 407 is biocompatible and FDA-approved for pharmaceutical applications, making it suitable for translational research. Additionally, its amphiphilic nature facilitates particle size control and homogeneous distribution during nanoprecipitation.This information jas been added to the manuscript.

6)    In page 4, please explain how the use of the xCELLigence RTCA DP system enhances the accuracy or real-time monitoring of ATD-PLGA NPs effect on CCA cell invasion compared to traditional endpoint assays.

Response: The xCELLigence RTCA DP system enhances accuracy by providing label-free, real-time monitoring of CCA cell invasion through continuous impedance-based measurements. Unlike traditional endpoint assays that capture only a single time point, this system tracks dynamic cellular responses to ATD-PLGA NPs over time, enabling precise assessment of the onset, duration, and magnitude of anti-invasive effects with improved reproducibility and sensitivity.This information jas been added to the manuscript.

7)    In page 6, how does the biphasic release profile of ATD-PLGA nanoparticles contribute to their therapeutic effectiveness compared to a single-phase or immediate-release formulation?

Response: The biphasic release profile of ATD-PLGA nanoparticles, featuring an initial rapid release followed by sustained drug release, enhances therapeutic effectiveness of the compound by ensuring early tumor exposure to atractylodin and maintaining prolonged cytotoxic levels over time. This contrasts with single-phase or immediate-release formulations, which may lead to short-lived efficacy, rapid clearance, and reduced tumor retention. This information jas been added to the manuscript.

8)    In page 7, please explain how the distinct kinetic pattern of ATD-PLGA NPs in inhibiting CL-6 cell migration and invasion reflects differences in their mechanisms of action?

Response: The distinct kinetic pattern of ATD-PLGA NPs, showing rapid and sustained inhibition of CL-6 cell migration and invasion, reflects their enhanced cellular uptake and prolonged intracellular release of atractylodin. This enables early and continuous disruption of motility-related signaling pathways, unlike free atractylodin or 5-FU, which act more slowly or indirectly due to differences in uptake, stability, and target specificity. This information has been added to the manuscript.

9)    In page 10, it should be explained how the reduced number of micronuclei in ATD-PLGA NP–treated cells compared to the MMC control indicates a lower genotoxic potential, and what are the implications for long-term safety in therapeutic applications.

Response:  The reduced number of micronuclei in ATD-PLGA NP–treated cells compared to the MMC control indicates a lower genotoxic potential, suggesting that the nanoparticles do not significantly damage DNA or interfere with chromosomal integrity. This implies greater long-term safety for therapeutic applications, as it reduces the risk of mutation-related side effects or secondary malignancies during prolonged use.This information jas been added to the manuscript.

10)    In page 11, please explain why treatment with ATD-PLGA NPs leads to a stronger dose-dependent downregulation of pro-tumorigenic genes and upregulation of IL-10 compared to conventional 5-FU.

Response: Treatment with ATD-PLGA NPs leads to stronger dose-dependent downregulation of pro-tumorigenic genes and upregulation of IL-10 compared to 5-FU due to the sustained release, enhanced cellular uptake, and multi-targeted action of atractylodin. Unlike 5-FU, which primarily targets DNA synthesis, ATD-PLGA NPs modulate inflammatory and metastatic signaling pathways more effectively, contributing to broader and more potent gene regulatory effects. This information jas been added to the manuscript.

11)    In page 13, it should be explained why encapsulating ATD into PLGA nanoparticles significantly enhanced its anticancer efficacy and selectivity against CCA cells compared to free ATD and 5-FU.

Response: Encapsulating ATD into PLGA nanoparticles significantly enhanced its anticancer efficacy and selectivity against CCA cells by improving ATD’s solubility, stability, and sustained release profile, which collectively increased its intracellular retention and tumor-specific accumulation. Unlike free ATD and 5-FU, the nanoparticle formulation enabled efficient cellular uptake, prolonged drug exposure, and targeted modulation of cancer-related pathways—resulting in stronger cytotoxicity toward CCA cells with reduced toxicity to normal cells. This information jas been added to the manuscript.

12)    The English of paper is acceptable.

Response: -

Reviewer 2 Report

Comments and Suggestions for Authors

1. There have been many papers published on the use of nanoparticles loaded with drugs for anti-tumor purposes. What is the innovation of this paper? It is recommended to add it in the introduction.
2. How did the weight of mice change during the anti-tumor experiment? It is recommended to add the corresponding data in the paper.
3. There is only one nanoparticle in Figure 1. It is recommended to add a TEM image containing more nanoparticles.
4. Add scale bars to Figure 3 and 5.

5. It's recommended to add some recent references. 

Author Response

SUMMARY OF RESPONSE TO REVIEWERS’ COMMENTS

Reviewer Comments:

Reviewer 2:

  1. There have been many papers published on the use of nanoparticles loaded with drugs for anti-tumor purposes. What is the innovation of this paper? It is recommended to add it in the introduction.

Response: This study provides a proof-of-concept for using nanoencapsulation to overcome pharmacological limitations of traditional herbal compounds and establishes ATD-PLGA NPs as a promising targeted therapy for a highly aggressive cancer with limited treatment options.

The innovation of this paper lies in the development and application of atractylodin-loaded PLGA nanoparticles (ATD-PLGA NPs) as a novel nanomedicine for the treatment of cholangiocarcinoma (CCA). Key innovations include: i) Novel drug delivery platform: This is the first study to encapsulate atractylodin, a poorly water-soluble and low-bioavailability compound derived from Atractylodes lancea, in FDA-approved PLGA nanoparticles to enhance its anticancer efficacy and pharmacokinetics. The nanoprecipitation method was optimized to achieve high encapsulation efficiency (83%) and sustained biphasic drug release (92% within 7 days)., ii) Superior efficacy over standard chemotherapy: ATD-PLGA NPs showed significantly lower ICâ‚…â‚€ values and higher selectivity indices against CCA cell lines (CL-6 and HuCCT-1) compared to both free ATD and 5-fluorouracil (5-FU). They also outperformed 5-FU in suppressing cell migration, invasion, and inducing apoptosis (up to 83% apoptotic cells)., iii) Multi-target antitumor mechanism: The nanoparticles modulated gene expression related to angiogenesis (VEGF), invasion (MMP-9, TGF-β), and inflammation (TNF-α, IL-6, and COX-2), while upregulating IL-10, an anti-inflammatory cytokine. These results demonstrate multifaceted anticancer activity, including anti-inflammatory, anti-metastatic, and pro-apoptotic effects., iv) In vivo efficacy in CCA xenograft model: In a nude mouse model, ATD-PLGA NPs produced dose-dependent tumor suppression, extended survival, and comparable or superior outcomes to 5-FU with lower systemic toxicity., and v) Reduced genotoxicity: Mutagenicity assays revealed that ATD-PLGA NPs exhibited lower genotoxic potential than mitomycin C, indicating better safety for potential clinical use.

  1. How did the weight of mice change during the anti-tumor experiment? It is recommended to add the corresponding data in the paper.

Response: The data on body weight of mice were added to the results section.

  1. There is only one nanoparticle in Figure 1. It is recommended to add a TEM image containing more nanoparticles.

Response: Figure 1 has been revised.

  1. Add scale bars to Figure 3 and 5.

Response: Figures 3 and 5 have been revised, incorporating the scale bar.

Figure 3

Figure 5

  1. It's recommended to add some recent references.

Response: More than seven recent references were added in the discussion section.

Round 2

Reviewer 1 Report

Comments and Suggestions for Authors

The revisions are satisfactory. The paper can be accepted in the current format.